# Strength Tests of Alloys for Fixed Structures in Dental Prosthetics

**DOI:** 10.3390/ma15103497

**Published:** 2022-05-13

**Authors:** Łukasz Bojko, Anna M. Ryniewicz, Wojciech Ryniewicz

**Affiliations:** 1Faculty of Mechanical Engineering and Robotics, AGH University of Science and Technology, 30 Mickiewicza Ave., 30-059 Krakow, Poland; ryniewic@agh.edu.pl; 2Department of Dental Prosthodontics and Orthodontics, Faculty of Medicine, Jagiellonian University Medical College, 4 Montelupich Street, 31-155 Krakow, Poland; wojciech@ryniewicz.pl

**Keywords:** dental prosthetics, supporting structures, tensile, compression, stiffness

## Abstract

The production of fixed prosthetic restorations requires strength identification in terms of cognition and the targeted clinical applications. The aim of the study is to evaluate the static strength in axial tensile and compression tests of titanium and cobalt alloys for the supporting foundations of crowns and bridges produced using Computer Aided Design and Manufacturing (CAD/CAM) technologies: Direct Metal Laser Sintering (DMLS) and milling. The test materials are samples of Ti6Al4V and CoCrMo alloys obtained using digital technologies and, for comparison purposes, CoCrMo samples from traditional casting. For the studied biomedical alloys, *R*_0.05_, *R_p_*_0.2_, *R_m_* and *R_u_* were determined in the tensile tests, and in the compression tests *R*_0.01_, *R_p_*_0.2_ and the stress σ at the adopted deformation threshold. Tensile and compression tests of titanium and cobalt alloys indicate differences in strength parameters resulting from the technology applied. The manufacturing of the structures by DMLS provides the highest stress values that condition elastic deformations for cobalt biomaterials: *R*_0.05_ = 1180 MPa, *R*_0.01_ = 1124 MPa and for titanium biomaterials: *R*_0.05_ = 984 MPa, *R*_0.01_ = 958 MPa. The high resistance to deformation of CoCrMo and Ti6Al4V from DMLS may be beneficial for fixed prosthetic structures subjected to biomechanical stresses in the stomatognathic system and the impact of these structures on the dento-alveolar complex.

## 1. Introduction

The construction of a fixed prosthesis must meet the anatomical and biomechanical conditions resulting from the function of the masticatory organ. They are determined using the analysis of occlusion, the state of muscle tension and work, and the dynamics of the articulation states of the mandible. Its functionality and aesthetics are determined by clinical procedures, manufacturing technology, the biomaterial used and its strength parameters [1,2,3,4,5,6,7,8].

An important area of research when using titanium and cobalt biomaterials from Computer Aided Design and Manufacturing (CAD/CAM) technologies for the supporting structures of crowns and bridges is the determination of their strength parameters [9,10,11,12]. In a stomatognathic system, these structures work in an elastic range and, due to veneering with ceramics, they must have appropriate stiffness. Clinical observations in studies indicate that, in the solutions of fixed dentures manufactured by traditional casting, the veneering ceramics were chipped off the substructure (Figure 1). The detachment of the ceramics most often occurs in the vicinity of the gingival area.

This zone is characterized by a thin veneer layer, proximity to the step, the formation of potential areas of plaque accumulation and the occurrence of maximum stress concentrations resulting from the occlusion and chewing function [13,14]. Based on clinical observations and previously determined biomechanical indications identifying the differentiation of stress distributions, displacements and deformations in the modeled prosthetic restorations of the masticatory organ, we believe that the strength parameters that characterize elastic deformations and protect the structure against plastic deformation should be determined. This analysis seems to be particularly important in the case of veneered layered structures, which must have appropriately limited values of elastic deformations in supporting structures. The preliminary tests performed showed that there are very significant differences in the strength of the same biomaterials, depending on the technology of producing the structure.

The aim of the research is to evaluate the static strength in axial tensile and compression tests of biomedical alloys on the supporting foundations of crowns and prosthetic bridges manufactured using technologies based on the CAD/CAM system. The conducted experiments will allow questions to be answered concerning the selection of a biomaterial with adequate strength in in vivo conditions. On the basis of the stress–strain characteristics, it is possible, inter alia, to indicate the stress values at which the sample will not undergo plastic deformation.

The study undertook strength tests of metal biomaterials for the production of prosthetic structures in two CAD/CAM technologies: Direct Metal Laser Sintering (DMLS) and milling. Both technologies are based on changing the manufacturing method by introducing new digital applications in clinical mapping, in modeling and in the technological process, which allows for better management of requirements. The production of structures in CAD/CAM procedures eliminates the analog method of mapping the prosthetic base. For comparison, the CoCrMo alloy was also tested for structures made in the traditional casting [15]. Traditional casting technology is still widely used today.

There is a lot of information available in the literature on individual technologies, concerning material compositions [16,17], manufacturing procedures [3,7,11,12,18], and microstructural and micromechanical analyses [1,9,10,19,20,21,22,23,24,25]. However, they often concern only one group of alloys. There is much less information and studies of extensive strength tests. The comparison of the resistance to elastic deformation and the determination of the yield strength of the biomaterial seems to be particularly important for the construction of prosthetic restorations in confrontation with biomechanical excitations. There is no information in the literature on comparing the strength parameters of these three technologies, especially in the area where the following technologies are used: DMLS, CAD/CAM milling and casting for the rehabilitation of the masticatory organs.

The study covers the presentation of the research material produced in the CAD/CAM procedures and in the casting method on professional dental devices and equipment. Tensile and compression tests were performed on a strength machine. Based on the stress–strain characteristics, the offset elastic limit, offset yield strength, ultimate tensile strength, breaking stress and compressive stress at a 15% strain were determined. Statistical analysis of the results allowed for their discussion, analysis and confirmation of the hypothesis. The publication ends with a discussion and conclusions.

## 2. Materials and Methods

The research material was samples of Ti6Al4V titanium alloy and CoCrMo cobalt alloy made of selective powders with the additive manufacturing technology in CAD/CAM. They were made of EOS Titanium Ti64 and EOS CobaltChrome SP2 powders, in the EOSINT M270 device from EOS. In the CAD/CAM milling process, factory discs made of Ti6Al4V alloy after casting and forging, Starbond Ti5 Disc (Grade 5) ELI Ti6Al4V and CoCrMo alloy after the sintering process of pressed powder, Starbond CoS Soft Disc, type 4, were used. Test samples were milled in CAD/CAM from fittings—discs on a 5-axis CORiTEC 350i dental milling machine by imes-icore. In addition, the group of comparative materials includes samples made of the CoCrMo type 5 alloy called Remanium GM 800+ by Dentaurum, made with the the lost-wax casting technology. The research program was approved by the Bioethics Committee of the Jagiellonian University (Approval No. KBET 122.6120.18.2016). Informed consent was obtained from all subjects involved in the study.

It was necessary to prepare specimens with specific dimensions, in accordance with the standards for tension and compression. The tensile samples had a measuring section with a constant section of 0.8 mm × 0.8 mm and were terminated with heads of increased dimensions (Figure 2). With an appropriate measuring length and a smooth transition to the heads, it can be assumed that the state of deformation and stress at each point of the measuring part is homogeneous. Under such conditions, it is possible to deduce the deformation of the samples and, from the measurements of the total force, the stress generated inside can be calculated. In the axial compressive strength tests, the dimensions of the samples had the shape of cuboids with a height of 3 mm and a square base of 2 mm × 2 mm. For each biomaterial used in the tensile and compression tests, 25 test sets were prepared. For the DMLS, 25 samples of both Ti6Al4V and CoCrMo were prepared, for CAD/CAM milling, 25 samples of both Ti6Al4V and CoCrMo and for the casting, and also 25 samples of CoCrMo.

Tensile and compression tests were carried out on an Instron 5566 testing machine, in accordance with the applicable ISO 6892-1:2019 standard [26]. A strain gauge head with a range of up to 10 kN was used and a constant strain rate of Vε = 10^−3^ s^−1^ was adopted. Grips were used for stretching, and the tests were carried out in a saline environment at 37 °C. A magnetic stirrer equipped with a heating plate was used, thanks to which a constant temperature was maintained in the entire volume of saline. The bath temperature corresponded to the conditions in the oral cavity. In the tests of materials, the value of the applied forces and the deformation caused by them were recorded, up to the destruction of the samples. Ceramic grips were used to stabilize the samples under compression (Figure 3). During the tests, the samples were compressed axially with a quasi-static load. The value of the applied forces and the deformations caused by them were recorded. The research was carried out to destruction. The compressive stresses σ were determined with the adopted deformation threshold of 15%.

## 3. Results

### 3.1. Tensile Test

For the tested biomaterials, stress–strain characteristics were prepared in tensile tests. On their basis, the offset elastic limit *R*_0.05_, offset yield strength *R_p_*_0.2_, ultimate tensile strength *R_m_* and breaking stress *R_u_* were determined. Statistical analysis of the test results was performed for all the samples. The study enabled the indication of mean values, standard deviations and scattering of the research results on the strength parameters of biomaterials (Table 1). An exemplary study for the determination of the offset yield strength *R_p_*_0.2_, depending on the production technology, has been included (Figure 4).

Based on the tensile tests of Ti6Al4V alloys from DMLS and CAD/CAM milling, differences in strength parameters depending on the manufacturing technology were observed (Table 1). The tensile characteristics of the Ti6Al4V alloy samples from both technologies differed. The tensile course of the Ti6Al4V samples from DMLS indicated a higher value of the offset elastic limit *R*_0.05_ by over 250 MPa, a higher value of the offset yield strength *R_p_*_0.2_ by about 310 MPa and the ultimate tensile strength *R_m_* by about 320 MPa. However, the samples were damaged at significantly lower percentages of deformation than in the case of samples from milling. This may show that the Ti6Al4V alloy from DMLS is stronger and more brittle than from CAD/CAM milling. By analyzing the applications of the studied biomaterials for crowns and bridges, it can be concluded that structures made of Ti6Al4V from sintering are characterized by greater resistance to deformation and greater strength than from milling.

The lowest resistance to deformation was found for CoCrMo from CAD/CAM milling, and the highest resistance to deformation, but with the highest brittleness, from DMLS. Photos of the samples before the tensile test and after their tearing confirmed different percentage deformations depending on the manufacturing technology (Figure 2). Statistical studies of the research results were performed for all samples. The study presents exemplary analyses of the normality distribution of the offset yield strength for the CoCrMo alloys from DMLS and casting (Figure 5 and Figure 6). In the tensile tests, the percentage difference in the average values of the strength parameters of the biomaterials from the analyzed technologies in relation to DMLS was also determined (Table 1).

### 3.2. Compressive Strength Tests

For the tested biomaterials, stress–strain characteristics were prepared in static axial compression tests. On their basis, the offset elastic limit *R*_0.01_, offset yield strength *R_p_*_0.2_ and compressive stress σ were determined at the assumed deformation threshold of 15%. The statistical analysis made it possible to indicate mean values, standard deviations and dispersions of the biomaterial strength test results (Table 2, Figure 7).

Compressive strength tests of the Ti6Al4V alloys from DMLS and CAD/CAM milling showed differences in strength parameters depending on the manufacturing technology (Table 2). The Ti6Al4V alloy from DMLS was characterized by an almost 180 MPa higher offset elastic limit *R*_0.01_ and 310 MPa higher offset yield strength *R_p_*_0.2_ than from CAD/CAM milling. The obtained graphs show the same inclination of the compression curves of the Ti6Al4V alloy from both technologies, which proves that the material was equally hardened during the test.

Tests of compression of the CoCrMo alloy from DMLS, CAD/CAM milling and traditional casting showed significant differences in strength parameters, depending on the manufacturing technology. The CoCrMo alloy from DMLS obtained the highest offset elastic limit *R*_0.01_—about 1120 MPa, higher by almost 600 MPa than in the case of milling and over 310 MPa higher than casting, as well as the highest offset yield strength *R_p_*_0.2_—about 1250 MPa, higher by over 680 MPa than in the case of milling and over 570 MPa higher than casting. Analyzing the slope of the compression curves of the CoCrMo alloy from all three technologies, it can be concluded that, during the compression test, the greatest hardening of the material occurs in the samples from casting, and slightly smaller, comparable to each other, in the samples from DMLS and milling.

The study presents exemplary analyses of the normality distribution of the offset yield strength for the Ti6Al4V alloys from DMLS and from milling (Figure 8 and Figure 9). In the compression tests, the percentage difference in the average values of the strength parameters of the biomaterials from the analyzed technologies in relation to DMLS was also determined (Table 2).

### 3.3. Statistical Analysis

The results were statistically analyzed using Statistica 13.3 (TIBCO Software Inc., Palo Alto, CA, USA). The following was designated:Descriptive statistics (mean, median, min, max, standard deviation);Normality of the distribution of variables (Shapiro–Wilk test, Kołmogorow–Smirnow test);Tests of the analysis of variance (ANOVA);Post-hoc multiple comparison test (Tukey, Bonferroni);

The level of statistical significance was assumed to be α = 0.05.

## 4. Overview and Discussion

Many research centers have dealt with the issues of tensile strength tests of Ti6Al4V alloys and CoCrMo alloys [16,17,18,19,20,21,22,23,24,27,28,29]. Yamanaka et al. [22,23,27] conducted tests of the tensile strength of the biomedical CoCrMo alloy after multi-pass thermomechanical treatment. Bereš et al. [21] conducted studies on the mechanical-phase behavior of the biomedical CoCrMo alloy. Vieira Muterlle [24] investigated the tensile strength of Ti6Al4V and CoCrMo alloys for biomedical applications. Mengucci et al. [16,18,19] investigated the tensile strength of Ti6Al4V (EOS Titanium Ti64) alloys and CoCrMo (EOS CobaltChrome SP2) alloys produced using DMLS. Kierzkowska [28] investigated the mechanical properties of the Ti6Al4V ELI alloy on the basis of a static tensile test. Kong et al. [20] conducted tests of the tensile strength of Ti6Al4V alloy samples from Selective Laser Melting (SLM) produced on a working platform in three directions of *x*, *y* and *z*. Dobrzański [29] conducted research on the influence of the technology of manufacturing microporous skeletons selectively laser sintered from titanium on the tensile strength.

In the tensile tests we performed, both for the Ti6Al4V alloy and CoCrMo alloy, the highest strength parameters were observed for laser sintering from selective metal powders. Cobalt alloys had strength parameters higher than those of titanium alloys. The exception is the cobalt alloy from DMLS, which under compression conditions is characterized by a slightly lower stress value at a strain of 15%. This situation also applies to the cobalt alloy from milling, which has the lowest value of compressive stresses at a strain of 15% of all the biomaterials tested. Cobalt alloys from traditional casting, in tensile and compression tests, are characterized by strength parameters that better protect against elastic deformation (at higher values of the offset elastic limit and the offset yield strength) than alloys from milling.

The highest values of stresses, which were provided by elastic strains—below the yield strength in tensile tests—were found for the CoCrMo alloys and Ti6Al4V alloys from DMLS. The lowest value of stresses provided by elastic strains was determined for the CoCrMo samples from milling. The CoCrMo alloys and Ti6Al4V alloys from DMLS were characterized by the highest tensile strength value. The lowest tensile strength was found for the CoCrMo alloy samples from milling.

Compressive strength tests of Ti6Al4V alloys and CoCrMo alloys have been carried out by many research centers [25,29,30,31,32]. Weißmann et al. [25] determined the compressive strength of scaffolds made of Ti6Al4V alloy from SLM. Nganbe et al. [30] performed compression tests of the head of a hip joint made of Ti6Al4V and CoCrMo alloys. Ramaswamy et al. [31] conducted tests of the compressive strength of Ti6Al4V alloy reinforced with yttrium oxide. Pang et al. [32] investigated the dynamic mechanical responses to compression of a selectively laser melted Ti6Al4V alloy. Dobrzański [29] conducted research on the influence of the technology of manufacturing microporous skeletons selectively laser sintered from titanium on the compressive strength.

In the compression tests carried out by our team, both for the Ti6Al4V alloy and CoCrMo alloy, the strength parameters of biomaterials were identified for laser sintering from selective metal powders, for CAD/CAM milling and for traditional casting. The compression tests showed the highest values of the offset elastic limit and the offset yield strength for the CoCrMo and Ti6Al4V alloys from DMLS. The lowest value of stresses provided by elastic strains was determined for the CoCrMo samples from milling. The highest value of compressive stresses at 15% strain was shown by the CoCrMo and Ti6Al4V alloys from DMLS. The CoCrMo alloys from milling had the lowest value of compressive stresses at 15% deformation, and those from casting had a higher value.

Tensile and compressive tests of titanium and cobalt alloys indicate differences in strength parameters resulting from the technology of biomaterials, with the same material composition. As shown above, the technology of manufacturing structures by sintering from metal powders has a fundamental impact on the strength parameters of biomaterials in the area of protection against elastic and plastic deformation. This regularity applies to CoCrMo alloys and, to a lesser extent, Ti6Al4V alloys.

To broaden the discussion regarding clinical applications, the results of the study on the longitudinal elasticity modulus and microhardness of the discussed biomaterials, determined by the nanoindentation method, are also presented (Table 3) [15,33]. They show that the technology of producing titanium and cobalt biomaterials does not cause any significant differences in the modulus of elasticity in the sintering and in the milling. Modulus of elasticity for titanium alloys is: for DMLS—112.5 GPa, and for milling—115.2 GPa, and for cobalt alloys: for DMLS—201.0 GPa and for milling—203.8 GPa. This may be due to the type and nature of the chemical bonds (bond stiffness) that are specific to certain titanium and cobalt alloys. However, it can be concluded that, when an object is characterized by a high value of the elasticity modulus, the stress effect is small—which is the case with CoCrMo alloys. In the case of titanium alloys, with almost half the value of the modulus of elasticity, the stress causes much greater strain and greater deformation of the object may occur. There are significant differences in microhardness between technologies and between biomaterials, which was identified on the basis of previous micromechanical studies [15,33]. In the case of titanium alloys, in the DMLS, the microhardness determined by the nanoindentation method was at a much higher level—4356.9 MPa than in the milling—3627.1 MPa. A similar nature of the relationship occurred for cobalt alloys. Microhardness was much higher for DMLS—6582.3 MPa than for milling—4951.0 MPa. On the other hand, in the casting, the microhardness was at the level of 5720.7 MPa, and the Young’s modulus was 193.2 GPa. The micromechanical tests showed similar regularities as in the tensile and compression tests, which confirm the results of our experiments. This is of particular importance in the assessment of biomaterials for the production of fixed prosthetic structures and the functional parameters that are required for them.

The tensile and compressive strength tests we presented of titanium alloys and cobalt alloys intended for application in dental prosthetics, obtained as a result of final procedures for the production of restorations on professional equipment, are innovative and have a clinical purpose. As previously mentioned—performed in other centers—strength tests provided valuable knowledge in the field of microstructure, micromechanical parameters, phase issues, single technologies (either sintering or milling), the influence of technology on the phase structure, or mechanical-phase behavior. Our research is focused on a specific application. They include a comparison of the technologies used in the production of fixed prosthetic structures: two computer-aided technologies and the traditional casting technology. For all technologies and for both types of alloys—titanium and cobalt, the basic strength parameters were determined, but special attention was paid to the offset elastic limit, unlike the tests that are available. Its determination seems to be extremely important for prosthetic restorations such as crowns and bridges made of metal alloys, which must be veneered with ceramics. It is the occurrence of elastic deformation, and then plastic deformation, which determines the functionality of the structure and its long-term prognosis. In addition to veneering, attention should be paid to the special features of these structures. Their anatomical and biomechanical shape must be preserved. When the permissible stresses are exceeded, the cross-sections cannot be increased—as in other types of structures. The prosthetist is obliged to follow the strict rules of saving the patient’s tissues. At the same time, the conditions of occlusion and chewing as well as subsequent tissue strain cannot be disturbed. On the basis of the presented strength tests, the doctor, taking into account all aspects of the patient’s supply, may have an indication to choose a biomaterial and the technology of manufacturing a prosthetic restoration. Of course, it must take into account a whole range of other conditions, including indications resulting from the geometric parameters of the structure—depending on whether it has a smaller or greater range, whether it is a prosthetic or implant-prosthetic structure, and what the optimal features of the veneering ceramics should be in terms of its adhesiveness to the supporting structure [7,12,34,35]. A prosthetic structure that is permanently deformed under chewing conditions was improperly made. The performed tests and their results allow for providing a qualitative strength indication for the clinical structures used subjected to biomechanical loads. In tensile and compression, the highest resistance to plastic deformation for the same geometry will be characterized in turn: cobalt structures produced by sintering, titanium structures made by sintering and milling, and then cobalt structures by casting and milling.

The stress values, which define the offset yield strength, are an indirect indication for adopting the criterion of strength assessment, because the functional safety of the layered structure must be ensured under the conditions of complex biomechanical loads.

The conducted strength tests allow the elastic and plastic properties for each biomaterial production technology to be checked and compared. Such knowledge, taking into account the clinical aspects, enables the optimal provision for the patient of a fixed prosthetic structure.

## 5. Conclusions

Based on the strength tests of biomaterials, including Ti6Al4V titanium alloys and CoCrMo cobalt alloys, on the supporting structures of prosthetic crowns and bridges, made using CAD/CAM and traditional systems, the technology has been found to have an influence on: the offset elastic limit and the offset yield strength in tension and compression, ultimate tensile strength, tensile stresses in tensile tests, and compressive stresses at a 15% strain.

The highest values of stresses characterizing: the elastic limit in tensile *R*_0.05_, the elastic limit in compression *R*_0.01_, and the yield strength in tension and compression *R_p_*_0.2_, below which the elastic deformations and stiffness of the structure are secured, were found in strength tests for the CoCrMo alloys and Ti6Al4V alloys from laser sintering. For cobalt alloys, they were as follows: *R*_0.05_ = 1180 MPa, *R*_0.01_ = 1124 MPa, in tension *R_p_*_0.2_ = 1225 MPa and in compression *R_p_*_0.2_ = 1246 MPa. For titanium alloys, the values were: *R*_0.05_ = 984 MPa, *R*_0.01_ = 958 MPa, with tension *R_p_*_0.2_ = 1046 MPa and with compression *R*_p0.2_ = 1168 MPa—intermediate for CoCrMo from traditional casting, and the lowest value was determined for the CoCrMo samples from milling.

The CoCrMo and Ti6Al4V samples from laser sintering were characterized by the highest value of ultimate tensile strength *R_m_*, and the highest value of breaking stresses *R_u_*. For cobalt alloys, their values were: *R_m_* = 1346 MPa, *R_u_* = 1332 MPa. For titanium alloys, they were: *R_m_* = 1112 MPa, *R_u_* = 989 MPa.

The highest value of compressive stresses at 15% strain was characteristic for samples from the laser sintering technology: CoCrMo—1619 MPa and Ti6Al4V—1641 MPa.

The CoCrMo alloy samples from milling had the lowest compressive strength at 15% deformation.

The high resistance to elastic deformation of CoCrMo and Ti6Al4V from laser sintering may be beneficial for fixed prosthetic structures subjected to biomechanical stresses in the SS and the impact of these structures on the dento-alveolar complex.

## Figures and Tables

**Figure 1 materials-15-03497-f001:**
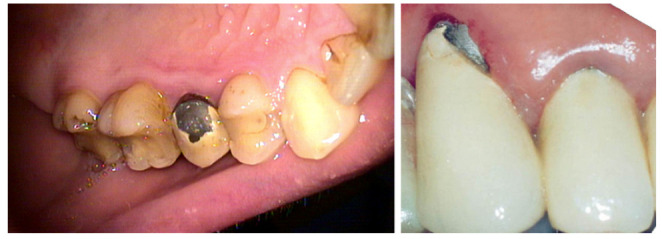
Detachment of veneering ceramics from CoCrMo cast crowns and veneered with dedicated ceramics.

**Figure 2 materials-15-03497-f002:**
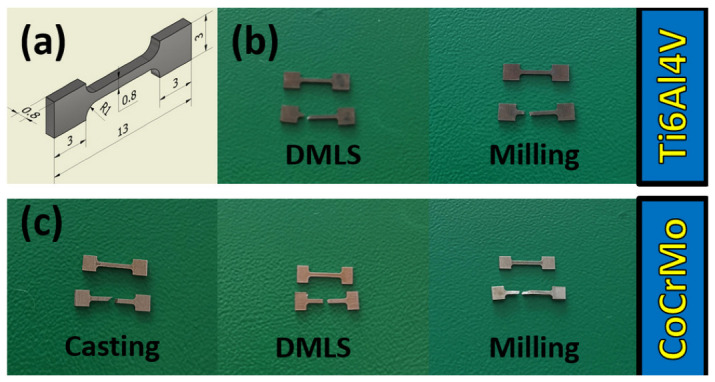
Material for tensile strength tests: (**a**) CAD model of the test sample; (**b**) Ti6Al4V alloy samples before and after the test; (**c**) CoCrMo alloy samples before and after the test.

**Figure 3 materials-15-03497-f003:**
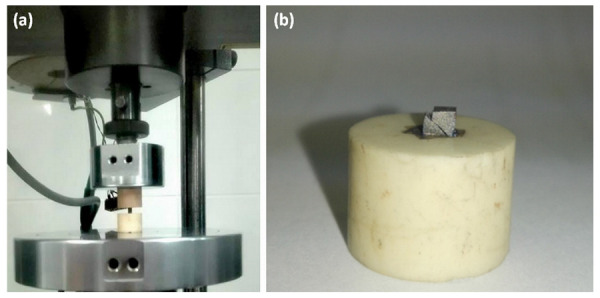
Compressive strength test: (**a**) test node during loading; (**b**) CoCrMo alloy sample from DMLS after the compression test.

**Figure 4 materials-15-03497-f004:**
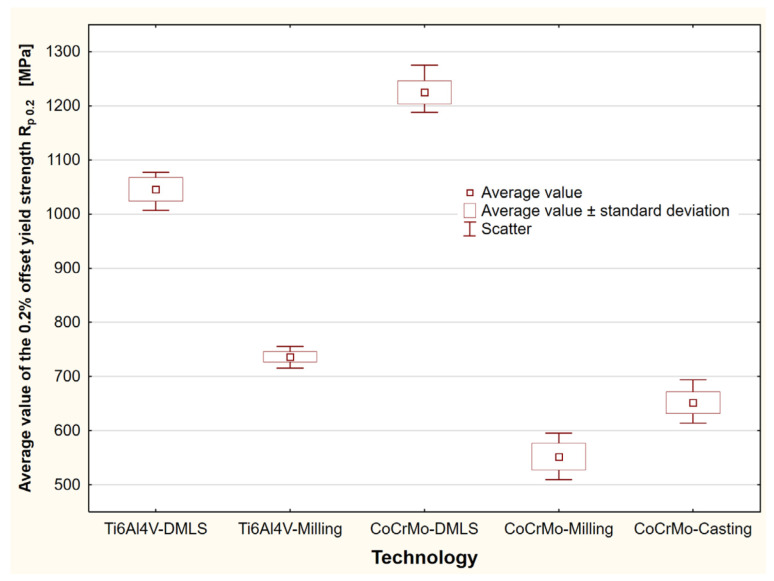
Statistical study to determine the offset yield strength under axial tension, depending on the manufacturing technology.

**Figure 5 materials-15-03497-f005:**
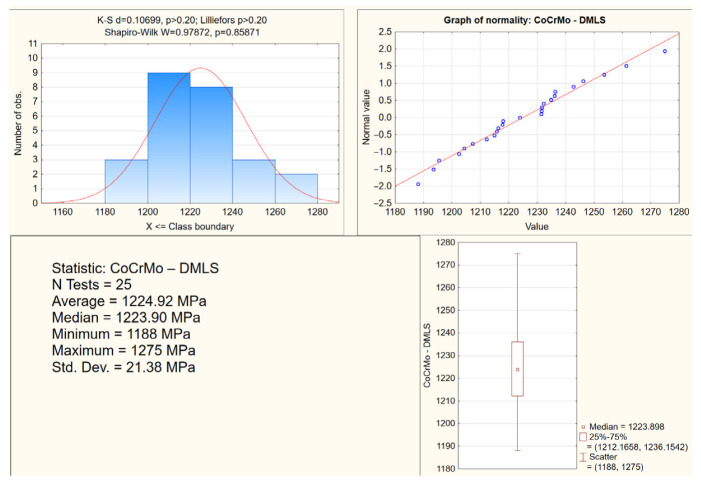
An exemplary analysis of the distribution normality of the offset yield strength in tensile tests for CoCrMo from DMLS.

**Figure 6 materials-15-03497-f006:**
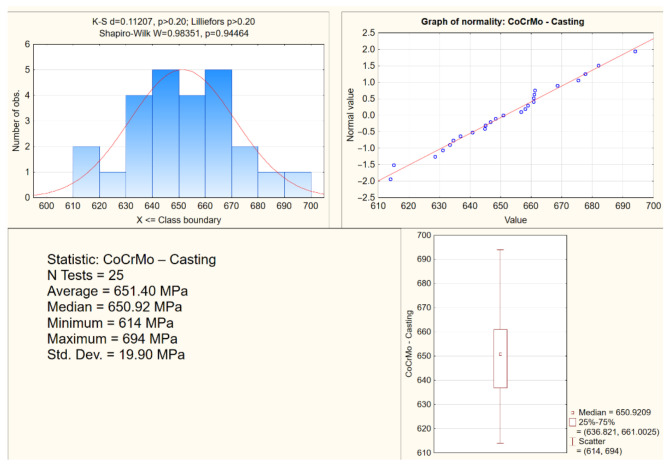
An exemplary analysis of the distribution normality of the offset yield strength in tensile tests for CoCrMo from casting.

**Figure 7 materials-15-03497-f007:**
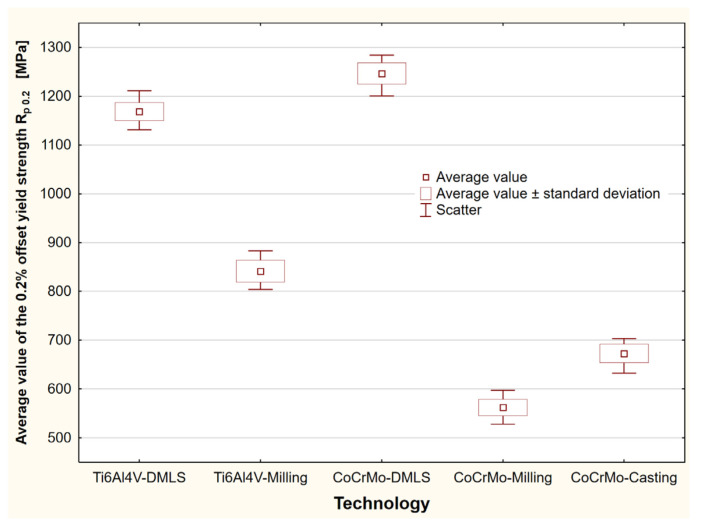
Statistical study to determine the offset yield strength under compression, depending on the manufacturing technology.

**Figure 8 materials-15-03497-f008:**
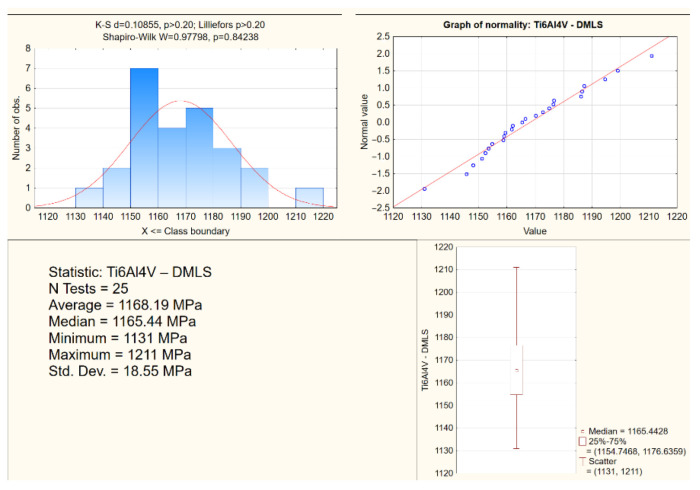
An exemplary analysis of the normality of the offset yield stress distribution in compression tests for Ti6Al4V from DMLS.

**Figure 9 materials-15-03497-f009:**
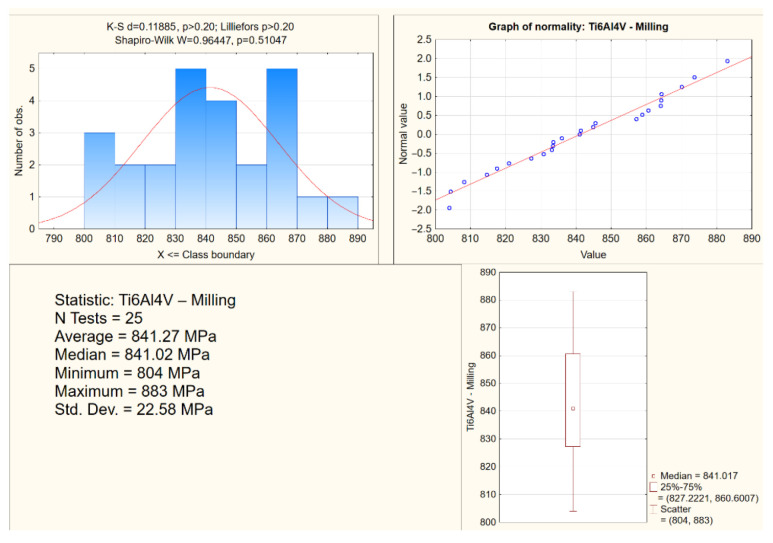
An exemplary analysis of the normality of the offset yield stress distribution in compression tests for Ti6Al4V from milling.

**Table 1 materials-15-03497-t001:** List of strength parameters of the tested biomaterials, depending on the production technology, in axial tensile tests.

Material	Technology	Metrological Parameter	Offset Elastic Limit *R*_0.05_ [MPa]	Offset Yield Strength *R_p_*_0.2_ [MPa]	Ultimate Tensile Strength *R_m_* [MPa]	Breaking Stress *R_u_* [MPa]
**Ti6Al4V**	**DMLS**	Average value	984	1046	1112	989
		Standard deviation	24.71	21.91	16.53	22.84
	**Milling**	Average value	731	736	796	779
		Standard deviation	13.39	9.81	11.54	25.44
	**The difference of average values of milling parameters in relation to DMLS, [%]**		25.71	29.64	28.42	21.23
**CoCrMo**	**DMLS**	Average value	1180	1225	1346	1332
		Standard deviation	19	21.38	23.17	29.52
	**Milling**	Average value	565	552	759	726
		Standard deviation	21.47	24.66	15.66	18.67
	**The difference of average values of milling parameters in relation to DMLS, [%]**		52.11	54.94	43.61	47.08
	**Casting**	Average value	646	651	794	717
		Standard deviation	16.83	19.90	28.93	37.89
	**The difference of average values of casting parameters in relation to DMLS, [%]**		45.25	46.86	43.61	47.74

**Table 2 materials-15-03497-t002:** List of strength parameters of the tested biomaterials, depending on the manufacturing technology, in axial compression tests.

Material	Technology	Metrological Parameter	Offset Elastic Limit *R*_0.01_ [MPa]	Offset Yield Strength *R_p_*_0.2_ [MPa]	Compressive Stress at 15% Strain *σ* [MPa]
**Ti6Al4V**	**DMLS**	Average value	958	1168	1641
		Standard deviation	23.97	18.55	37.78
	**Milling**	Average value	779	841	1151
		Standard deviation	27.49	22.58	41.36
	**The difference of average values of milling parameters in relation to DMLS, [%]**		18.68	28.00	29.86
**CoCrMo**	**DMLS**	Average value	1124	1246	1619
		Standard deviation	26.55	22.09	51.93
	**Milling**	Average value	532	562	897
		Standard deviation	20.03	16.81	49.12
	**The difference of average values of milling parameters in relation to DMLS, [%]**		52.67	54.90	44.60
	**Casting**	Average value	580	673	1176
		Standard deviation	21.51	18.98	35.62
	**The difference of average values of casting parameters in relation to DMLS, [%]**		48.40	45.99	27.36

**Table 3 materials-15-03497-t003:** The results of the modulus of longitudinal elasticity and microhardness determined by the nanoindentation method [15,33].

Material	Technology	Metrological Parameter	Young’s Modulus *E* [GPa]	Microhardness HiT [MPa]
**Ti6Al4V**	**DMLS**	Average value	112.5	4356.9
		Standard deviation	3.5	184.1
	**Milling**	Average value	115.22	3627.1
		Standard deviation	11.0	382.2
**CoCrMo**	**DMLS**	Average value	201.0	6582.3
		Standard deviation	11.2	452.4
	**Milling**	Average value	203.8	4951
		Standard deviation	17.7	218.8
	**Casting**	Average value	193.2	5720.7
		Standard deviation	8.4	362.7

## Data Availability

The data presented in this study are available on request from the corresponding author. The data are not publicly available due to it being part of a large database and linked to other clinically proprietary personal data.

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
