# Peer review of "Strength Tests of Alloys for Fixed Structures in Dental Prosthetics"

_materials, 2022, doi:10.3390/ma15103497_

Round 1

Reviewer 1 Report

the authors made a good effort in mechanical analysis of biomaterials, however, attention to the following points is required.

A) what is your hypothesis?
B) no gap of literature is mentioned! what are you going to answer based on the literature.? is your study brand new?

C) line 45: the sentence is incomplete!

D) line 52-62: this paragraph may need to be summarized or shifted to the discussion section.

E) the standard, study, and control groups need to be identified clearly in the M&M section.
F) line 197-209: this paragraph should be revised. it is only the reports of the previous study without analysis or comparison!
G) line 235-243: the sole report of previous studies should be avoided. your paragraph should compare and contrast the available related data!
H) minor grammar revision and formatting correction are required. the long sentence should be cut into two or three easily readable ones.

best

Reviewer 2 Report

The authors present an experimental study on the mechanical tensile and compression testing for the determination of the strength of, materials, used for prosthetic restorations. The research work is performed carefully by selecting the materials as Ti6Al4V and CoCrMo, and they are manufactured by DMLS as well as traditional methods. This article requires the following major corrections to be made to be considered for publication 

1. "Digital Technologies" - Why did the authors choose the word digital technologies? In general, the term digital technologies refer to complete computational methods. DMLS, which falls under 3D printing, may be digital, but the term digital technologies and this term in the title are confusing. Please correct it throughout the article. 
2. The introduction seems to be short, and does not explain the complete need for the work, and lacks few other literatures which have done in the past. I notice that these articles are described in the "Discussion" section, however, little other information can be included in the introduction 

3. For this short introduction, too many articles are cited, and most of them appear to be self-citation of the authors 

4. Even though CAD/CAM is a common term, in a technical paper, it is recommended to include a full abbreviation for its 1st appearance in the article. 

5. At the end of the introduction, please include an outline of the article, on how the remaining of this article is arranged. 

6. How are the samples prepared/manufactured ? More specifics can be included, included the equipment details, so that work can be reproduced or compared by other researchers

7. Line 77, where 25 samples prepared by each methodology or altogether ? 

8. The Test Standard ISO 6892-1: 2016-09 is an expired standard. The current one is 2019. Why was the new standard not followed ? Also please do cite the standards. 

9. In the results, you have provided a table and plot. It would also be nice to include the % difference of the average values of strength, between the 2 manufacturing methods. Repeat for both tensile and compression results. 

10. In the overview and discussion section, you have mentioned the literature work done by other researchers. What are the limitations of those work, and why did you have to do the experiments and report this research ? What is lacking in the industry ? 

Reviewer 3 Report

Comments to authors are listed below:

  • The abstract lacks to present the numerical values from significant findings in this paper.

  • Regarding the discussion of results, no new insight has been made, and limitation in the tests of this manuscript to make an impact based on the results presented.

  • The manuscript does not contain enough experimental data and it can’t have a scientific impact. The mechanical tests are not enough to concluded about the novelty applications of this work.

  • The discussion of results is tremendously poor and  brief based on the limitation in the tests of the manuscript.

Unfortunately, I cannot recommend the manuscript for publication.

Round 2

Reviewer 1 Report

After Considering other reviewers' comments, the paper can proceeds for publication. 

Author Response

Thank you for recommending the article for publication

Reviewer 2 Report

Thank you for making all the corrections, and providing clear explanations to the questions. The article is now recommended for publication

Author Response

(The authors gave the same response as above.)

Reviewer 3 Report

Comments to authors

  • The discussion is still required more explanations of the results presented in this paper.
  • The conclusions should include the significant numerical finding from the results obtained in this work.
